# VUGEN: Visual Understanding priors for GENeration

## ABSTRACT

Recent advances in Vision-Language Models (VLMs) have enabled unified understanding across text and images, yet equipping these models with robust image generation capabilities remains challenging. Existing approaches often rely on reconstruction-oriented autoencoders or complex bridging mechanisms, leading to misalignment between understanding and generation representations, or architectural complexity. In this work, we propose VUGEN, a novel framework that explicitly leverages VLM's pretrained visual understanding priors for efficient and high-quality image generation. Our approach first transforms the high-dimensional latent space of the VLM's native vision encoder into a lower-dimensional, tractable distribution that maximally preserves visual information. The VLM is then trained to sample within this reduced latent space, ensuring alignment with its visual understanding capabilities. Finally, a dedicated pixel decoder maps these generated latents back to the image space. We find that a VAE-free pixel diffusion decoder to be on par or better than commonly used complex latent diffusion decoders that internally rely on VAE latents. Extensive experiments demonstrate that VUGEN achieves superior image generation performance, improving DPG Bench from 71.17 to 74.32 and FID from 11.86 to 9.06 on COCO, while fully preserving the VLM's original understanding capabilities.

## 1 INTRODUCTION

Recent years have witnessed a remarkable evolution in Large Language Models (LLMs), which now demonstrate impressive capabilities in both understanding and generating natural language. Building on this progress, Vision-Language Models (VLMs) have emerged, extending the capabilities of LLMs by incorporating visual understanding. This advancement enables unified reasoning across both text and images. As research continues to push the boundaries, there is growing interest in unified VLMs—or Multimodal LLMs (MLLMs)—that can accept both text and images as input and output, thereby combining understanding and generation capabilities across modalities.

A central question arises from this trajectory: Given that VLMs are pretrained to develop rich visual priors for understanding, can these learned priors be effectively harnessed for image generation? Our preliminary experiments suggest that the vision embeddings produced by a VLM's vision encoder —despite being optimized for understanding—retain sufficient visual information to support high-quality image generation. This observation motivates our core research question: *How can we efficiently leverage the visual understanding priors learned by VLMs to enable generation?*

Existing approaches to unifying vision and generation in VLMs have notable limitations. Recent methods (Wu et al., 2025a; Deng et al., 2025; Liao et al., 2025) adopt semantic vision encoders (*e.g.* SigLIP (Zhai et al., 2023)) for image understanding, and introduce image tokenizers from reconstruction-oriented autoencoders (*e.g.* (VQ-)VAE (Kingma & Welling, 2014; Van Den Oord et al., 2017)) to handle image generation. While this decoupled approach enables flexible integration of specialized experts, it induces a misalignment between the representations used for understanding and generation, hindering the model's ability to fully leverage shared semantic information. Another line of work seeks to bridge VLMs with image diffusion models through various connection mechanisms. These methods (Gupta et al., 2022; Chen et al., 2025a; Lin et al., 2025) teach VLMs to sample semantic representations from a pretrained semantic image encoder. However, these approaches tend to be architecturally complex, increasing overall model size and computational footprint. More-

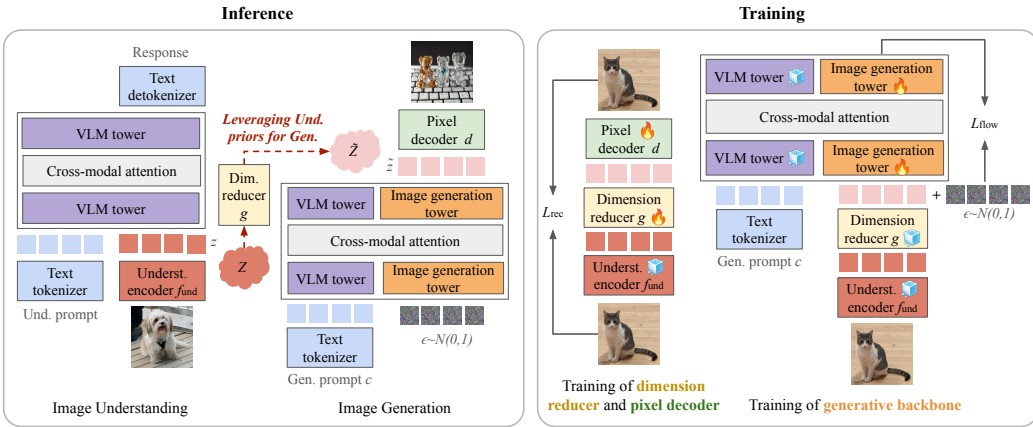

Figure 1: **VUGEN inference (left):** The complex VLM vision encoder space $\mathcal{Z}$ is reduced in dimension to $\tilde{\mathcal{Z}}$ for generative modeling. VUGEN samples in $\tilde{\mathcal{Z}}$, and the pixel decoder maps the generated latents to image space. **VUGEN training (right):** We first jointly train the dimension reducer and pixel decoder to ensure a latent space optimized for generation. Then the learned dimension reducer is frozen, and the VLM is trained to sample over the (fixed) reduced space $\tilde{\mathcal{Z}}$.

over, directly generating in the high-dimensional semantic representation space is challenging and necessitates large models to capture the complex distributions involved.

In this work, we propose **VUGEN**: **V**isual **U**nderstanding priors for **GEN**eration to directly addresses these challenges. It decomposes image generation into two stages: (i) The VLM backbone learns to sample in the latent space of its native understanding vision encoder. This design allows us to fully leverage the VLM's pretrained vision priors and avoids misalignment introduced by decoupled tokenizers. However, the semantic understanding embeddings are typically high-dimensional and complex to model. We therefore introduce a dimension reducer that simplifies these embeddings while preserving the essential information for high-quality image generation, and the VLM learns to sample in the reduced space. This makes the generative modeling tractable and enables efficient training with smaller models. (ii) A pixel decoder then maps the generated latents back to pixel space. We explore two approaches for this stage: (a) finetuning a pre-trained text-to-image latent diffusion model that conditions on the understanding latents, and (b) training a lightweight, VAE-free pixel diffusion decoder that directly reconstructs images from the understanding latents. This eliminates the architectural complexity and VAE dependence common in prior work. Experimental results demonstrate that our method significantly enhances prompt following capabilities and image generation quality: VUGEN improves the baseline DPG Score (Hu et al., 2024) from 71.17 to 74.32 and FID from 11.86 to 9.06 on COCO2014 dataset (Lin et al., 2014).

Our contributions can be summarized as follows:
- VUGEN uses the VLM's inherent understanding embedding space as the intermediate representation for image generation. By combining this with a novel dimension reduction technique, we effectively and efficiently transfer the VLM's understanding priors to its generative capabilities.
- We find that a direct pixel-diffusion decoder improves over latent diffusion model decoders while eliminating the dependence on VAE-based tokenizers and reducing architectural complexity.
- We conduct extensive experiments on two datasets of varying scale, demonstrating both qualitatively and quantitatively that VUGEN improves image generation performance while preserving the base VLM's understanding abilities.

## 2 RELATED WORK

**Unified vision language models.** Early examples of such models, such as Chameleon (Chameleon Team, 2024), Show-O (Xie et al., 2025), Transfusion (Zhou et al., 2025) and Emu3 (Wang et al., 2024), relied on image tokenizers derived from reconstruction-oriented discrete or continuous autoencoders like (VQ-)VAE (Kingma & Welling, 2014; Van Den Oord et al., 2017). The multimodal model then generates vision tokens either autoregressively or via diffusion, depending on the discrete or continuous nature of the tokenizer. However, reconstructive autoencoders like VAEs are known

to be suboptimal for visual understanding tasks, limiting their effectiveness in this context. Aiming to combine strong understanding performance of VLMs with generative capabilities, another line of work, including Janus (Wu et al., 2025a), JanusFlow (Ma et al., 2025), Bagel (Deng et al., 2025), and Mogao (Liao et al., 2025) leverage separate semantic encoders for understanding and VAEs for generation. While this decoupling can improve individual task performance, the use of distinct encoders can cause representational inconsistencies between understanding and generation pathways. As a result, models must adapt to a new encoding space, leading to lower training efficiency (Deng et al., 2025). More recent approaches such as Metamorph (Gupta et al., 2022), MetaQueries (Pan et al., 2025), BLIP3-o (Chen et al., 2025a) and Bifrost (Lin et al., 2025), generate continuous visual representations from LLMs/VLMs to guide an external latent diffusion model. This, however, significantly increases the overall model size and computational footprint, and does not eliminate the dependence on VAE latent spaces since the diffusion model itself typically relies on it. In contrast to these different prior approaches, our method relies on a single visual representation that is optimized for and supports strong image understanding performance (rather than (VQ-)VAEs in early work), while using a simple pixel diffusion decoder that is light-weight in parameters and compute (rather than relying on repurposed text-to-image latent diffusion models).

**Two-stage generative image models.** Several prior works have explored two-stage image generation pipelines for text-to-image and unconditional image generation. A first (diffusion) model is trained to generate a high-level semantic representation, *e.g.*, CLIP latents (Ramesh et al., 2022), SSL features (Li et al., 2024), or features from pretrained image classifier (Pernias et al., 2024). A second (diffusion) model then takes these (generated) high-level features as input and models the conditional distribution over images, either directly in pixel-space (Ramesh et al., 2022) or over a VAE latent space (Li et al., 2024; Pernias et al., 2024) which is itself decoded to pixel-space using the VAE decoder. In a similar spirit, we learn a stage-one text-conditional diffusion model over semantic VLM features, and generated features are decoded to pixel space using a second-stage diffusion model. Different from these prior works, our approach seamlessly instills image generation capabilities on top of pretrained VLMs using its native vision features. REPA (Yu et al., 2025) is an alternative approach to leverage high-level features for generation, where internal representations of the generative model are aligned with pre-trained image embeddings, rather than decomposing the generative process to explicitly generate such features as an intermediate step. In our experiments we consider this approach as a baseline.

**Pixel-space diffusion decoders.** While latent diffusion models (LDMs) (Rombach et al., 2022) underpin many state-of-the-art generative image models, recent work has shown that pixel-space diffusion models (Hoogeboom et al., 2025) can achieve competitive image quality and faster training. Pixel-space diffusion models have also been adopted as decoders in autoencoder frameworks, enabling more expressive decoders that can accurately model the conditional distribution over images given latent representations (Zhao et al., 2025; Chen et al., 2025c). These decoders have demonstrated improved image reconstruction and generation, particularly at larger spatial downsampling factors. This is well-suited for visual understanding features, which typically use $16\times$ downsampling. In our work, we leverage a pixel-space diffusion model to decode the (generated) VLM understanding features. Our design enables a fully unified visual representation for both understanding and generation, while at the same time reducing the model complexity and computational overhead associated with LDM-based decoders.

## 3 VUGEN: VISUAL UNDERSTANDING PRIORS FOR GENERATION

VUGEN equips a pretrained Vision-Language Model (VLM) with text-to-image generation capabilities by fully leveraging its learned visual understanding priors, while also retaining its original understanding abilities. We aim to utilize the VLM's native image understanding embeddings as an (intermediate) generative target, thereby aligning the generation process with the VLM's pretrained semantic priors and visual comprehension. However, directly generating in the full embedding space $\mathcal{Z}$ of the VLM's vision encoder is not trivial due to is high dimensionality and complexity. To make the generation process tractable, we introduce a **dimension reducer** $g$ that morphs the native embedding space into a more tractable lower-dimensional space $\tilde{\mathcal{Z}}$, while preserving the essential information required for recovering high-quality visual content.

Given an image $x \in \mathcal{X}$ and a corresponding textual prompt $c$, VUGEN's text-to-image generation process is decomposed into two stages. **Generative modeling**: We model the conditional distribution $P(\tilde{z}|c)$, *i.e.* we train the VLM to sample from the reduced understanding space $\tilde{\mathcal{Z}}$. **Pixel decoding**: A dedicated pixel decoder $d$ maps the generated $\tilde{z}$ back to the image space, modeling $P(x|\tilde{z})$. Overall, the generation process can be factorized as: $P(x|c) = P(\tilde{z}|c) \cdot P(x|\tilde{z})$. An overview of VUGEN's inference and training process is illustrated in Figure 1. In the remainder of this section we detail the dimension reducer in Section 3.1, the generative modeling on understanding features in Section 3, and the pixel decoder in Section 3.3.

## 3.1 DIMENSION REDUCTION OF UNDERSTANDING EMBEDDING SPACE

For visual understanding, VLMs employ a semantic image encoder $f_{\text{und}}$ to project images $x$ into semantic embeddings $z$. These embeddings are then mapped into a shared embedding dimension and fused with text features via self-attention in the transformer backbone. Although $f_{\text{und}}$ is optimized for understanding tasks, we observe that $\mathcal{Z}$ captures a remarkably rich signal of visual appearance. As demonstrated in Figure 2(a,b), images can be effectively reconstructed solely from their understanding embeddings $z$. This proves that $z$ retains sufficient visual signal for image generation. Moreover, $\mathcal{Z}$ is an ideal target for generative modeling: if we train the VLM to sample

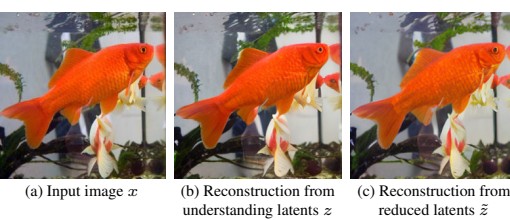

(a) Input image $x$    (b) Reconstruction from understanding latents $z$    (c) Reconstruction from reduced latents $\tilde{z}$

Figure 2: Images reconstructed from understanding latents. Accurate reconstruction indicates that both the full $z$ and the reduced $\tilde{z}$ understanding latents retain sufficient visual information.

directly over $\mathcal{Z}$, we ensure that the produced latents are inherently compatible with the VLM's own semantic space—allowing us to fully exploit its pretrained visual priors and achieve seamless integration between understanding and generation.

However, in practice, we find that directly training a generative model over $\mathcal{Z}$ is difficult, with FID of generated samples above 200. We hypothesize that this is because $\mathcal{Z}$ exhibits a complex distribution, especially when compared to the compact and structured latent spaces produced by VAE-based approaches commonly used in existing methods. VAEs are explicitly trained to facilitate reconstruction, encouraging the model to compress information into a dense, lower-dimensional manifold (typically $8\times$ spatially downsampled with 4 channels) that is easier to model and sample from. In contrast, each $z \in \mathcal{Z}$ in our setting is formed by concatenating the understanding features of all image patches, resulting in a much sparser and significantly higher-dimensional representation (e.g., $14\times$ downsampling but 1024 channels for the vision encoder used in our experiments).

To overcome this challenge, we introduce a dimension reducer $g$ that projects the high-dimensional embedding space $\mathcal{Z} \subset \mathbb{R}^D$ into a lower-dimensional, more tractable space $\tilde{\mathcal{Z}} \subset \mathbb{R}^{\frac{D}{r}}$, where $D$ is the dimension of understanding embeddings and $r$ denotes $g$'s reduction ratio. Formally, $\tilde{z} = g(z) = g \circ f_{\text{und}}(x)$. Then we learn a generative model on $\tilde{\mathcal{Z}}$ instead of directly on $\mathcal{Z}$. Abstractly, $g$ should extract the most relevant features for generation from the high-dimensional embeddings and discard redundancy. In this way, $g$ bridges the gap between the rich, high-dimensional representations learned by the VLM and the practical requirements of generative modeling.

A simple approach to compress $z$ is to use linear dimension reduction techniques such as PCA, which retains the linear subspace spanning most of the signal variance. In practice, however, images reconstructed from $\tilde{z}_{\text{PCA}} = W_{\text{PCA}}^\top z$ lack crucial visual details. This indicates a critical misalignment between the features that account for the most variance and those that are essential to capture visual content for generation. To address this, we propose to jointly learn the dimension reducer with the stage-two pixel decoder by parameterizing the dimension reducer as a trainable module $g_\phi$ of the pixel decoder $d_\psi$. This joint learning enables the reducer to dynamically extract latent features from the understanding encoder that are maximally suitable for high-fidelity image generation, see Figure 2(c).

## 3.2 LEARNING TO SAMPLE IN REDUCED UNDERSTANDING SPACE

With the reduced latent space $\tilde{\mathcal{Z}}$ established, we train our generative model to sample in this space, fully leveraging the VLM's visual priors while ensuring tractability. Concretely, we adopt rectified flow matching (Liu et al., 2023; Lipman et al., 2023). Let $t \in [0, 1]$ interpolating between the compressed encoding $\tilde{z} = g \circ f_{\text{und}}(x)$ of an image $x \in \mathcal{X}$ and unit Gaussian noise $\epsilon \sim \mathcal{N}(0, 1)$, where we denote the interpolant as $\tilde{z}_t = t\tilde{z} + (1 - t)\epsilon$. For training we use the flow matching loss:

$$L_{\text{flow}}(\theta) = \mathbb{E}_{t, x \sim \mathcal{X}, \epsilon \sim \mathcal{N}(0,1)} \parallel \tilde{z} - \epsilon - v_{\text{VLM},\theta}(\tilde{z}_t, t|c) \parallel^2, \tag{1}$$

where $v_{\text{VLM},\theta}$ denotes the VLM image generation tower that acts as velocity field predictor. Note that, since our goal is to sample from the fixed, tractable distribution $\tilde{\mathcal{Z}}$, both the dimension reducer $g$ and the understanding encoder $f_{\text{und}}$ are kept frozen during this training stage.

To best preserve the base VLM's pretrained visual and language knowledge, we adopt the Mixture of Transformer (MoT) architecture (Liang et al., 2025). Specifically, we initialize a new trainable image generation tower from the pretrained VLM weights, and we keep the main VLM tower frozen during training. During text-to-image generation, cross-modal interactions are facilitated through cross-modal attention: following the masking strategy of Transfusion (Zhou et al., 2025), we apply causal attention to text tokens and bidirectional attention to all vision tokens. In line with the flow matching paradigm, inference proceeds iteratively: we start by appending pure noise $\tilde{z}_T \sim \mathcal{N}(0, 1) \in \tilde{\mathcal{Z}}$ to the input sequence of the text prompt, the model predicts the velocity field at each step $t$, which is then integrated to update the current noisy latent $\tilde{z}_t$.

## 3.3 MAPPING FROM UNDERSTANDING LATENTS TO PIXEL SPACE

Given an input image $x$, the pixel decoder $d_\psi$ is trained to reconstruct the original image from its reduced image embedding $\tilde{z} = g_\phi \circ f_{\text{und}}(x)$. Formally, $\hat{x} = d_\psi(\tilde{z})$. As detailed in Sec. 3.1, the dimension reducer $g_\phi$ and decoder $d_\psi$ are jointly optimized. Importantly, this joint training is performed prior to training the VLM to sample in $\tilde{\mathcal{Z}}$. Once trained, the dimension reducer $g_\phi$ is kept frozen during the generative training stage. This ensures that the VLM learns to generate within a fixed, tractable latent space. We consider two pixel decoder designs.

**Latent Diffusion Model (LDM).** Similar to prior work on extending LLMs with image generation capabilities (Gupta et al., 2022; Pan et al., 2025), we consider leveraging a pre-trained text-to-image LDM. We replace the original conditioning on a sequence of text tokens with conditioning on $\tilde{z}$: a sequence of (reduced) understanding embeddings of image patches, and finetune the LDM to exploit this new conditioning signal. We follow the standard LDM training procedures, where the objective is to denoise a latent variable conditioned on the understanding embedding.

**Pixel-space Diffusion Decoder (PDD).** Alternatively, we consider a pixel-space diffusion decoder (PDD) that greatly simplifies the overall design by eliminating the dependence on a large LDM which itself internally uses a VAE latent space. We use an autoencoding formulation to train the PDD, where the understanding encoder $f_{\text{und}}$ serves as the frozen encoder, complemented with the trainable dimension reducer $g_\phi$, and the pixel decoder $d_\psi$ is trained to map the reduced embedding $\tilde{z}$ directly to pixel images. Our decoder is inspired by recent advances in diffusion autoencoders (Zhao et al., 2025; Chen et al., 2025c), which allow for more accurate reconstructions than classic feed-forward decoders due to the iterative denoising process.

## 4 EXPERIMENTS

### 4.1 EXPERIMENTAL SETUP

**Implementation details.** We initialize our Vision-Language Model (VLM) from Perception Language Model 1B (Cho et al., 2025). Following the Mixture of Transformers (Liang et al., 2025) approach, we initialize a new trainable image generation tower to process generative visual embeddings and keep the original VLM tower frozen. For the dimension reduction module $g_\phi$, we instantiate it as an MLP with SiLU (Elfwing et al., 2018) activation. We adopt the dimension reduction ratio $r = 16$ as our default setup. For the LDM decoder, we adopt a multi-modal DiT architecture (Esser

Table 1: Image generation results with VUGEN and baselines on StockMix and ImageNet.

| Model | StockMix | | | | ImageNet | | | |
|---|---|---|---|---|---|---|---|---|
| | FID ↓ | CLIP↑ | DPG ↑ | GenEval↑ | FID ↓ | CLIP↑ | Density↑ | Coverage↑ |
| Decoupled | 13.06 | 26.71 | 72.09 | 54.00 | 5.85 | 26.32 | 70.51 | 16.61 |
| REPA | 11.86 | 26.71 | 71.17 | 55.03 | 5.40 | 25.89 | 72.12 | 17.14 |
| VUGEN | **9.07** | **27.45** | **74.15** | **56.81** | **4.15** | **26.40** | **103.32** | **22.46** |

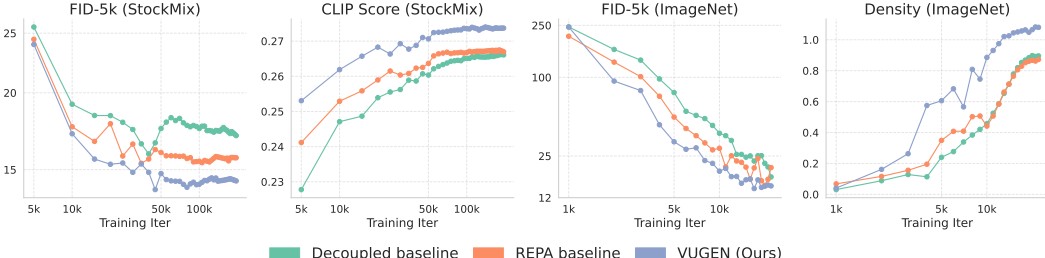

Figure 3: **Generation performance across training iteration on StockMix and ImageNet.** VU-GEN reaches better performance in fewer training steps than compared baselines.

et al., 2024), conditioning on (reduced) understanding latents via an adapted embedding layer. For the pixel diffusion decoder (PDD), we use a U-ViT-based architecture (Hoogeboom et al., 2023) with transformer blocks, trained with flow-matching (Lipman et al., 2023) and perceptual losses. See the App. A for more implementation details.

**Datasets.** We train on two datasets of varying scale and distribution. (i) **ImageNet-1k** (Deng et al., 2009) is widely used to benchmark image generation. For models trained on ImageNet, evaluation is performed on its official validation set. We follow the ImageNet preprocessing in standard diffusion literature (Karras et al., 2022). We use the prompt template *"This is an image of a [CLS]."* for conditioning on ImageNet class labels. (ii) **StockMix**: our primary training setup, composed of a mixture of YFCC100M (Thomee et al., 2016), CC12M (Changpinyo et al., 2021), and S320M—a large proprietary collection of stock images. This larger-scale dataset covers a more diverse real-world imagery. For models trained on StockMix, evaluation is conducted on the COCO2014 (Lin et al., 2014) validation set. For CC12M and S320M, we recaption images using Florence-2 (Xiao et al., 2024) to obtain higher-quality captions. All training and evaluation are performed at a resolution of $256 \times 256$ pixels.

**Metrics.** We evaluate image generation quality using FID (Heusel et al., 2017). To assess prompt alignment we report CLIP Score (Hessel et al., 2021), DPG-Bench (Hu et al., 2024) and GenEval (Ghosh et al., 2023). We include distributional metrics density and coverage (Naeem et al., 2020) to separately assess image quality and diversity.

**Baselines.** We compare against two main baselines using the same architecture, data and training setup to allow for apples-to-apples comparisons. (1) **Decoupled Vision Encoders Baseline.** We initialize from the same VLM backbone and use the VAE from SD3 (Esser et al., 2024) for generation. In other words, instead of sampling in the understanding embedding space, the model learns to sample VAE tokens, which are then decoded by the VAE decoder. Conceptually, this baseline closely resembles the LLaVAFusion setup from Shi et al. (2024), but differs in the architecture details and training data. (2) **REPA Baseline.** We extend our first baseline by adding REPA (Yu et al., 2025) to align the intermediate representations of the image generation tower with the understanding embeddings of input images from the native encoder $f_{\text{und}}$. This baseline is particularly relevant to our proposed approach, as it induces an *implicit* alignment with understanding embeddings, whereas VUGEN *explicitly* samples in the understanding latent space.

### 4.2 MAIN RESULTS

**Comparison with baselines.** We compare the image generation performance of VUGEN and the baselines in Tab. 1. We observe that VUGEN consistently outperforms baselines across both

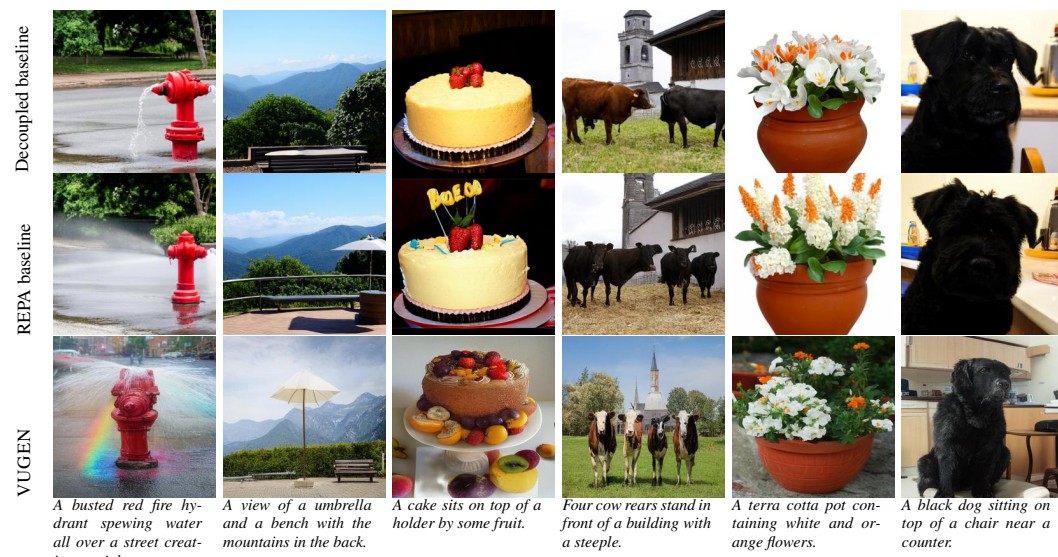

Figure 4: **Qualitative comparison of images generated by models trained on StockMix.** VU-GEN demonstrates significantly stronger prompt following capabilities (*e.g.* rainbow in column 1, umbrella and bench in column 2, fruit in column 3) and produces more realistic outputs as well as finer visual details (columns 4, 5 and 6).

datasets and on every reported metric. Specifically, VUGEN achieves a 24% improvement in FID on StockMix and 23% on ImageNet w.r.t. the REPA baseline, indicating that the generated images are closer to the real data distribution. The results also validate that generating in the understanding embedding space enhances semantic alignment, as evidenced by improvements in both dense prompt following (DPG-Bench) and fine-grained compositional evaluation (GenEval). On ImageNet, VUGEN also delivers substantial gains in density and coverage, highlighting its ability to produce images with both higher fidelity and greater diversity. For these experiments, we fix the classifier-free guidance (CFG) scale at 1.8 for ImageNet and 5.0 for StockMix to achieve the best balance between image quality and alignment. In App. D we provide analysis of the CFG strength.

In Fig. 3 we further extend the comparison between VUGEN and baselines to intermediate check-points, where we evaluate using 5,000 validation images. The results demonstrate that VUGEN achieves consistent improvement throughout training, and also reaches comparable performance much faster than the baselines. For example, on ImageNet, after 10k training steps VUGEN already matches the density of the REPA baseline at 30k steps. Additionally, by leveraging native vision embeddings as the target space for generation, VUGEN benefits from a stronger initialization. The prompt alignment (CLIP) score starts a high value since the beginning of training, indicating more effective and semantically meaningful learning from the outset.

**Qualitative results.** Qualitative comparisons in Fig. 4 present side-by-side examples of images generated by VUGEN and the baselines. VUGEN demonstrates significantly better prompt alignment and is able to accurately address semantic concepts that other methods fail to capture. Additionally, VUGEN produces images with finer details and overall higher visual fidelity. Furthermore, Fig. 5 examines the diversity of generated samples under the same prompt. While VAE-based baselines tend to produce images with similar object appearances and simple backgrounds, VUGEN generates a broader variety of samples, capturing richer variations in both content and style.

**Comparison with SOTA methods.** In addition to baseline comparisons, we compare VUGEN against SOTA unified VLMs that are capable of both image understanding and generation. Tab. 2 shows that VUGEN achieves state-of-the-art COCO FID among models of similar size and delivers competitive results on prompt alignment metrics of GenEval and DPG-Bench.

### 4.3 ABLATION STUDIES

**PCA reducer vs. jointly trained reducer.** We compare PCA-based dimension reduction with learning-based reducers using the LDM pixel decoder in Figure 6. In addition we consider the PDD

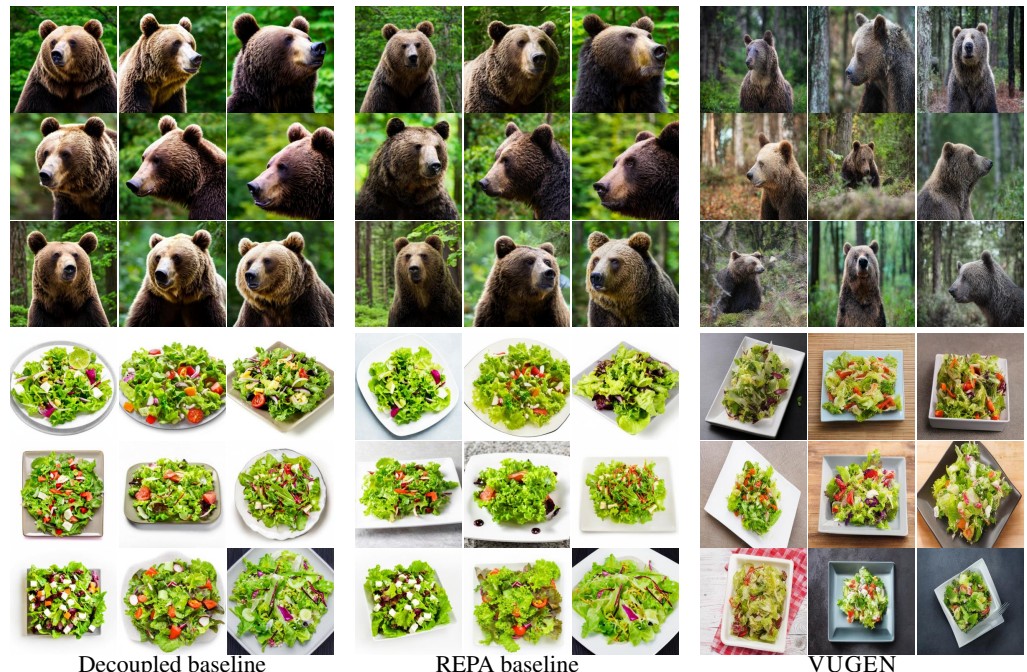

Decoupled baseline                REPA baseline                VUGEN

Figure 5: **Sample diversity analysis of models trained on StockMix.** Images generated with: *"A bear looking forward in a forest."* and *"A fresh looking salad on a square plate."*. While baselines tend to produce repetitive outputs (the same pose of the bear and uniform background for the salad), VUGEN exhibits variability in camera angles, backgrounds, and overall appearance.

| Method | Base (M)LLM | COCO FID↓ | GenEval↑ | DPG↑ |
|---|---|---|---|---|
| **7B+ scale** | | | | |
| EMU (Wang et al., 2024) | LLaMA 13B | 11.66 | - | - |
| MetaMorph (Gupta et al., 2022) | LLaMA-3 8B | 11.8 | - | - |
| TokenFlow-XL (Qu et al., 2025) | Qwen-2.5 14B | - | 0.63 | 73.38 |
| LMFusion (Shi et al., 2024) | LLaVA-Next 8B | 8.20 | - | - |
| DreamLLM (Dong et al., 2024) | Vicuna 7B | 8.46 | - | - |
| Chameleon (Chameleon Team, 2024) | From Scratch 7B | 26.74 | 0.39 | - |
| EMU3 (Wang et al., 2024) | From Scratch 7B | 12.80 | 0.66 | 80.60 |
| MetaQuery-XL (Pan et al., 2025) | Qwen2.5-VL 7B | 8.69 | 0.80 | 82.05 |
| JanusPro-7B (Chen et al., 2025b) | DeepSeek-LLM 7B | - | 0.80 | 84.19 |
| BLIP3-o 8B (Chen et al., 2025a) | Qwen2.5-VL 7B | - | 0.84 | 81.60 |
| Bifrost-1 (Lin et al., 2025) | Qwen2.5-VL 7B | 34.35 | 0.81 | 77.67 |
| **3B scale** | | | | |
| MetaQuery-L (Pan et al., 2025) | Qwen2.5-VL 3B | 8.87 | 0.78 | 81.10 |
| BLIP3-o 4B (Chen et al., 2025a) | Qwen2.5-VL 3B | - | 0.81 | 79.36 |
| Bifrost-1 (Lin et al., 2025) | Qwen2.5-VL 3B | 23.02 | 0.61 | 76.41 |
| **∼1B scale** | | | | |
| Show-o-512 (Xie et al., 2025) | Phi-1.5 1.3B | 9.24 | 0.68 | - |
| Janus (Wu et al., 2025a) | DeepSeek-LLM 1.5B | 8.53 | 0.61 | - |
| JanusFlow (Ma et al., 2025) | DeepSeek-LLM 1.5B | - | 0.63 | 80.09 |
| JanusPro-1B (Chen et al., 2025b) | DeepSeek-LLM 1.5B | - | 0.73 | 82.63 |
| VUGEN (CFG scale 5) | PLM 1B | 9.07 | 0.57 | 74.15 |
| VUGEN (optimal CFG scale / metric) | PLM 1B | 6.77 | 0.61 | 76.97 |

Table 2: Comparison with SOTA unified VLMs on image generation metrics. For VUGEN, we report metrics both at a fixed default guidance scale of 5 and at their respective optimal guidance scales, since fidelity and prompt alignment metrics peak at different values. We found the optimal guidance scale to be 1.5 for FID, 12 for GenEval, and 14 for DGP Score.

pixel decoder with a jointly learned reducer. The qualitative results show that images recovered from PCA latents are noticeably less accurate (see, *e.g.*, the dog tongue and teddy bear face), showing that high variance directions in understanding features are not per se the best for generation. We also compare the reconstruction FID curves during training of the LDM decoder with PCA and jointly trained reducer in the right panel of Figure 6. The jointly learned reducer maintaining a significant

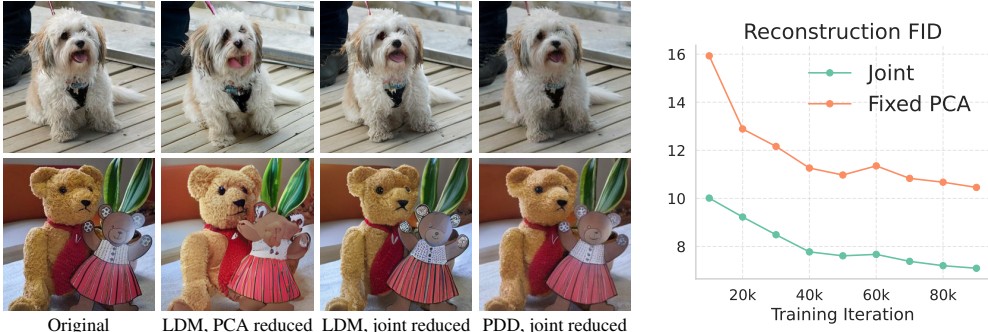

| Original | LDM, PCA reduced | LDM, joint reduced | PDD, joint reduced |

Figure 6: **Left:** Image reconstructions from reduced understanding latents ($r = 16$), using PCA and jointly trained reducers, using LDM and PDD pixel decoders. Reconstructions from PCA are more lossy (see, *e.g.*, the dog tongue and the teddy bear face). **Right:** Reconstruction quality over the course of training of LDM decoder with fixed PCA reducer vs. jointly learned reducer.

advantage throughout training over the PCA alternative, and we therefore retain the jointly trained reducer. Comparing the PDD and LDM decoders, we find them to achieve similar results.

**Dimension reduction ratio.** We examine the effect of the dimension reduction ratio $r$ by training models with $r = 1, 2, \ldots, 32$ on ImageNet. In Fig. 7 we consider the reconstruction and generation quality using the LDM decoder at different $r$. For reconstruction, the image quality deteriorates as $r$ increases, as there is less information in the reduced latent to recover the originals. For generation on the other hand, for $r < 4$ the generative model is not able to properly learn the complex high-dimensional distribution, and the FID remains very high. With larger reduction factors the model is able to fit the data distribution in reduced dimensions. These re-

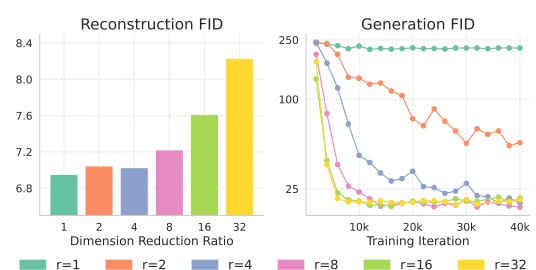

Figure 7: Influence of dimension reduction ratio $r$ on image reconstruction and generation quality.

sults highlight a tradeoff: a higher reduction ratio simplifies the generation over the reduced understanding features, while making the pixel decoding task more challenging. Optimal performance is achieved by balancing these competing factors, and in practice we set $r = 16$.

**Latent diffusion model decoder (LDM) vs. pixel-space diffusion decoder (PDD).** The qualitative results in Fig. 6 show that PDD and LDM produce reconstructions of similar visual fidelity. Despite being conceptually simpler, the PDD decoder is much more efficient both in terms of number of parameters (48M vs. 794M), and in terms of throughput (119.2 vs. 3.2 ims/sec with batch size 64 on a single H200 GPU). PDD is therefore a compelling alternative to the LDM decoders used in prior work, enabling image generation in multimodal models over understanding embeddings, and obviating the need for large LDM decoders that internally rely on reconstruction-based VAE latents.

## 5 CONCLUSION

In this work, we introduced VUGEN, a model that unifies visual understanding and generation within VLMs by leveraging their native understanding priors. By decomposing image generation into a tractable two-stage process —sampling in a reduced semantic latent space and decoding to pixels— we address key limitations of prior approaches, such as representation misalignment and architectural complexity. To the best of our knowledge, VUGEN is the first multimodal generative model that generates images without relying on any reconstruction-based autoencoder latent space. Our experiments demonstrate that VUGEN is competitive with state-of-the-art models, and achieves significant improvements in both prompt following and image quality w.r.t. strong baselines. These results highlight the effectiveness of directly harnessing VLM understanding embeddings for generation, paving the way for more integrated and efficient multimodal generative models.

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

## A  ADDITIONAL IMPLEMENTATION DETAILS

**Pre-trained VLM.** We base our models off the models distributed in the public PLM repository (Cho et al., 2025). Our VLM backbone is Perception Language Model 1B, which adopts Perception Encoder (Bolya et al., 2025) as its native understanding encoder and Llama 3 (Dubey et al., 2024) as the base language model. With the mixture of transformer setup, the total number of trainable parameters in VUGEN is 1.2B.

**Latent Diffusion Model (LDM) decoder.** We follow the LDM setup of Berrada Ifriqi et al. (2024), utilizing a multi-modal DiT architecture (Esser et al., 2024) with 28 blocks and a hidden size of 1152. The model employs the asymmetric autoencoder from Wu et al. (2023) to define its latent space. Training is performed under the EDM (Karras et al., 2022) formulation for the DDPM paradigm, with noise rescheduling as in Berrada Ifriqi et al. (2024), and sampling is conducted using the EDM Euler scheduler. We pretrain the LDM using the StockMix dataset. To enable decoding of (reduced) understanding latents, we adapt the condition embedding layer's input dimension to match the understanding latents, which are then fed into the DiT attention layers.

**Pixel Diffusion Decoder (PDD).** We use the pixel diffusion decoder from Anonymous (2026), which is based on a U-ViT architecture (Hoogeboom et al., 2023) with 3 downsampling stages and transformer blocks only at the deepest levels ($8 \times 8$ blocks). The model is trained with a flow matching loss (Lipman et al., 2023), LPIPS (Zhang et al., 2018), and REPA loss (Yu et al., 2025) for internal feature alignment with DINOv2-B features (Oquab et al., 2024). For fast decoding, the model is distilled into a single-step diffusion decoder (SSDD) similarly to Luhman & Luhman (2021), but keeping the full training loss, and using an 8-steps model as the teacher, a copy of the weights as the student. To decode from pretrained PLM understanding latents, we treat the PLM encoder as a frozen encoder and connect it to the decoder via a jointly learned dimension reducer. Full details of the pixel diffusion decoder can be found in (Anonymous, 2026).

**Training.** During training, we use a batch size of 8 and a sequence length of 4096, with a learning rate of $3 \times 10^{-4}$. We adopt an AdamW (Loshchilov & Hutter, 2019) optimizer with $\beta_1 = 0.9$, $\beta_2 = 0.95$, and weight decay of 0.1. We train for 200k iterations on 32 GPUs for the model trained on StockMix, and 50k iterations on 8 GPUs for ImageNet. All experiments are conducted on NVIDIA H200 GPUs. Exponential Moving Average (EMA) of weights is activated starting at 50k training iteration for StockMix and 10k for ImageNet, with a decay of 0.9999.

## B    COMPARISON WITH SOTA METHODS ON IMAGE UNDERSTANDING

Tab. 3 compares VUGEN's understanding performance with SOTA methods. By construction, VU-GEN retains the image understanding capabilities of the underlying base VLM, *i.e.* that of PLM-1B in our experiments. The results show that VUGEN has understanding performance that is comparable to that of other models at similar scale.

Table 3: Comparison with SOTA models on image understanding metrics

| Method | Base (M)LLM | MME-P↑ | MMB↑ | MMMU↑ |
|---|---|---|---|---|
| **7B+ scale** | | | | |
| MetaMorph (Gupta et al., 2022) | LLaMA-3 8B | - | 75.2 | - |
| LMFusion (Shi et al., 2024) | LLaVA-Next 8B | 1603.7 | 72.1 | 41.7 |
| TokenFlow-XL (Qu et al., 2025) | Qwen-2.5 14B | 1551.1 | 76.8 | 43.2 |
| BILP3-o 8B (Chen et al., 2025a) | Qwen2.5-VL 7B | 1682.6 | 83.5 | 50.6 |
| Bifrost-1 (Lin et al., 2025) | Qwen2.5-VL 7B | 1685.2 | 83.5 | 58.6 |
| MetaQuery-XL (Pan et al., 2025) | Qwen2.5-VL 7B | 1685.2 | 83.5 | 58.6 |
| JanusPro-7B (Chen et al., 2025b) | DeepSeek-LLM 7B | 1567.1 | 79.2 | 41.0 |
| Chameleon (Chameleon Team, 2024) | From Scratch 7B | - | - | 22.4 |
| VILA-U (Wu et al., 2025b) | LLaMA-2 7B | 1401.8 | - | - |
| EMU3 (Wang et al., 2024) | From Scratch 7B | - | 58.5 | 31.6 |
| **3B scale** | | | | |
| MetaQuery-L (Pan et al., 2025) | Qwen2.5-VL 3B | 1574.3 | 78.6 | 53.1 |
| BLIP3-o 4B (Chen et al., 2025a) | Qwen2.5-VL 3B | 1527.7 | 78.6 | 46.6 |
| **~1B scale** | | | | |
| Show-o-512 (Xie et al., 2025) | Phi-1.5 1.3B | 1097.2 | - | 26.7 |
| Janus (Wu et al., 2025a) | DeepSeek-LLM 1.5B | 1338.0 | 69.4 | 30.5 |
| JanusFlow (Ma et al., 2025) | DeepSeek-LLM 1.5B | 1333.1 | 74.9 | 29.3 |
| JanusPro-1B (Chen et al., 2025b) | DeepSeek-LLM 1.5B | 1444.0 | 75.5 | 36.3 |
| VUGEN | PLM 1B | 1546.2 | 75.8 | 32.1 |

## C    MORE QUALITATIVE RESULTS

We provide additional qualitative comparison on StockMix in Fig. 8. We also show examples of generated images by VUGEN trained on ImageNet in Fig. 9.

## D    REALISM-CONSISTENCY TRADE-OFF

In text-to-image generation models, there is usually a trade-off between consistency (*i.e.* prompt alignment) and realism (*i.e.* quality of generated samples). The CFG (Classifier-Free Guidance) scale serves as a crucial control parameter: higher values generally improve prompt alignment (consistency) but often at the expense of image realism. To analyze this trade-off, we sweep the CFG scale and evaluate the performance of different models using the CLIPScore (as a measure of consistency) and FID (as a measure of realism). We visualize these dynamics in Fig. 10 by plotting CLIP score and FID as functions of the guidance scale, as well as plotting CLIPScore vs. FID, where curves closer to the top-left indicate a more favorable balance between consistency and realism.

On the ImageNet dataset, our method consistently outperforms the baselines across the entire range of guidance scales, achieving both better CLIP and FID scores for every setting. This superiority is clearly reflected in the right-most plot, where our method's curve remains closer to the top-left, indicating a better consistency-realism trade-off. On StockMix, the comparison is more nuanced. While our method consistently achieves higher consistency scores across all guidance scales, it does not achieve the absolute lowest FID. At higher guidance scales, however, our method maintains a significantly lower FID than both baselines, indicating better realism when strong prompt consistency is required. Furthermore, the Pareto front across FID and ClipScore mostly consists of points from our method, except for the guidance scale 1.5 result where the baselines obtain better FID at the expense of worse CLIPScore.

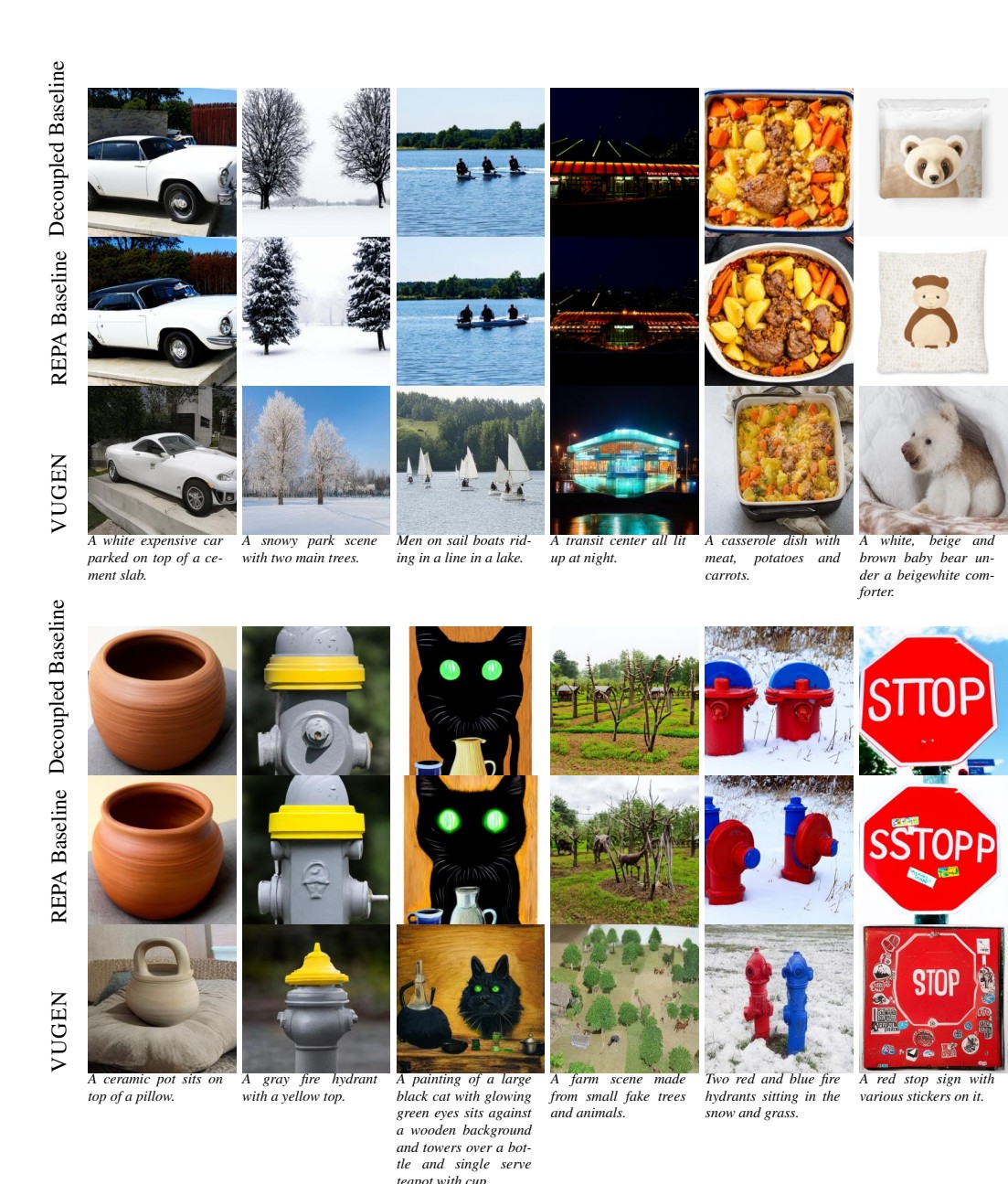

Figure 8: Qualitative comparison of VUGEN and baselines trained on StockMix.

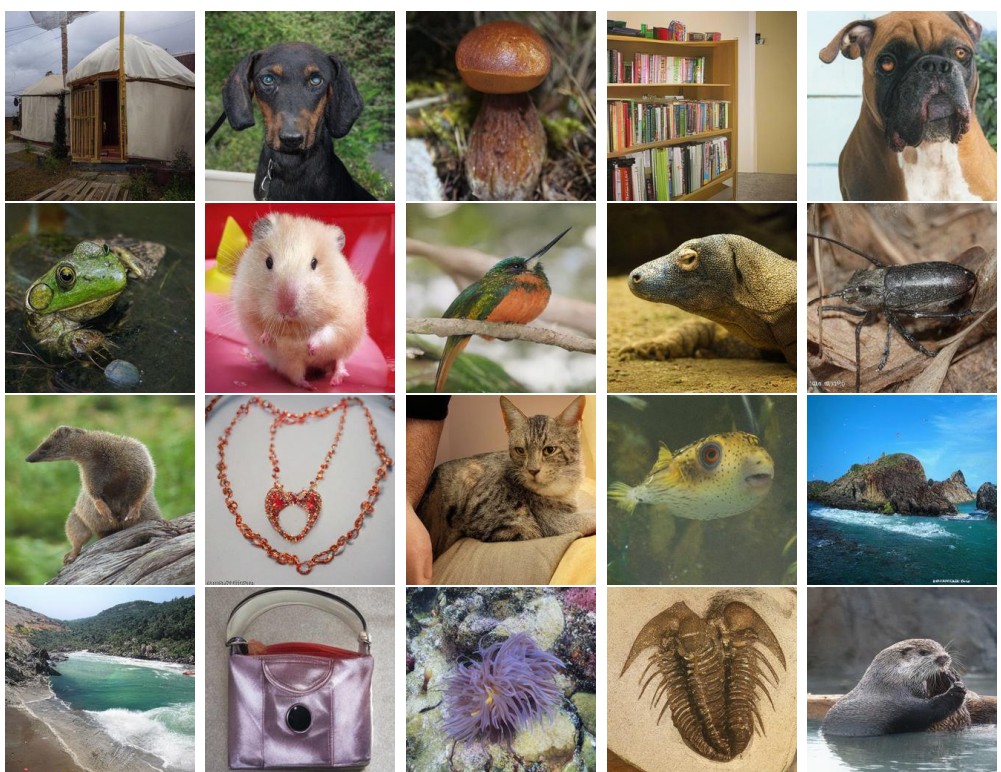

Figure 9: Qualitative results of VUGEN trained on ImageNet.

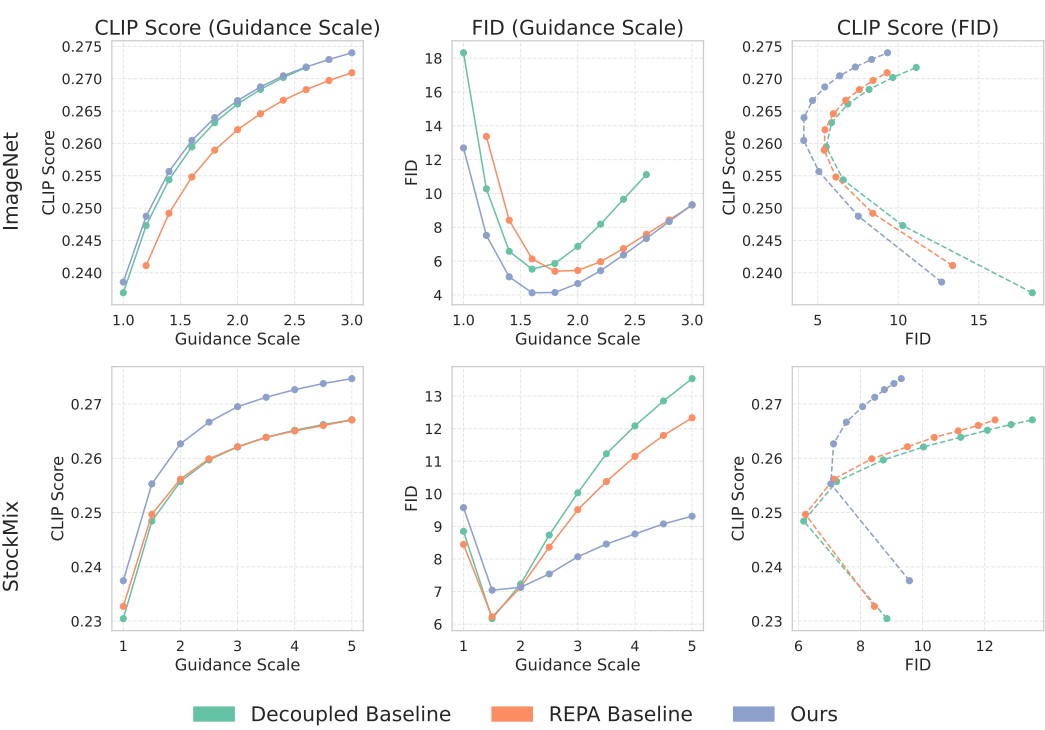

Figure 10: Consistency-realism trade-off analysis: CLIPScore (consistency) and FID (realism) are plotted as against classifier-free guidance (CFG) scale, along with CLIPScore vs. FID plot.

