# OpenReview forum: "VUGEN: Visual Understanding priors for GENeration"
_ICLR.cc/2026/Conference — Submitted to ICLR 2026_

### Official Review · Reviewer_7WAU · 2025-10-24

**Soundness:** 3
**Presentation:** 2
**Contribution:** 2
**Rating:** 2
**Confidence:** 3

**Summary:**

This paper proposes VUGEN, a framework that leverages pretrained visual understanding embeddings (from a frozen VLM) as priors for image generation. A learnable dimension reducer is introduced to map the high-dimensional understanding space to a lower-dimensional latent space, making it easier to model with a generative flow-matching network. The model then decodes the latent space into images using either a lightweight pixel diffusion decoder (PDD) or a latent diffusion decoder (LDM). Experiments on StockMix and ImageNet show that VUGEN outperforms VLM-based generation baselines.

**Strengths:**

The idea of reusing pretrained visual understanding embeddings as generative priors is both intuitive and meaningful, offering potential to bridge understanding and generation tasks in multimodal modeling. The use of a learnable dimension reducer to create a smoother and more compact latent space is technically sound and empirically validated. Additionally, the lightweight and fast PDD provides a practical alternative to standard latent diffusion decoders.

**Weaknesses:**

- This work primarily focuses on generation, with an emphasis on training and evaluating on generation tasks. Therefore, it should be compared to state-of-the-art generative models, rather than unified multimodal models (UMMs). Additionally, even when compared to UMMs, the generation performance of VUGEN is not particularly outstanding.

- If the paper aims to argue for VUGEN as a unified multimodal model (UMM), it falls short in terms of unification. Also, evaluations on visual understanding and reasoning tasks are necessary to fully justify its claim as a UMM.

- The core idea of the framework is to leverage understanding priors for generation, so further clarification and more analyses are necessary regarding the contribution of these priors, theoretically, qualitatively and quantitatively.

- The motivation is not clearly articulated enough in the abstract and introduction. The claim that generating in the understanding latent space is challenging requires more theoretical and empirical analyses.

- Clarifying the computational cost and scalability of the two-stage training would facilitate the cost-benefit comparisons.

**Questions:**

Please see weaknesses.

---

> ### Author Response · Authors · 2025-11-20
> **Response to reviewer's questions**
>
> We thank the reviewer for their detailed feedback. We address each concern below:
>
> **Weakness 1: VUGEN focuses on generation (…), it should be compared to generative models, rather than UMMs.**
>
> VUGEN’s primary contribution is enabling unified multimodal models (UMMs) from a pre-trained VLM. The goal is to achieve strong generation performance while maintaining the VLM's original understanding capabilities. A direct comparison with purely generative models would be inappropriate for the following reasons:
>
> - Data Setup: Our comparison is a controlled study of generative performance under the same data regime (Table 1). State-of-the-art generative models train on different data, have different scales, and use different architectures. Comparing directly would conflate these factors and make it harder to isolate our core contribution.
> - Architectural Constraints: VUGEN operates under the constraint of using VLM embeddings, not arbitrary latent spaces optimized solely for generation. Therefore, the appropriate comparison is against other models that also balance understanding and generation—i.e., other UMMs.
>
> That said, our results are competitive with state-of-the-art UMMs on generative tasks (Table 2, showing the best COCO FID amongst 1B-scale models). Additionally, models like Janus demonstrate that UMMs can be competitive with generative-only models like SDXL and Emu3-Gen on benchmarks like GenEval.
>
> **Weakness 2: Missing visual understanding and reasoning evaluation.**
>
> Table S1 in the Appendix presents evaluation results on visual understanding tasks, demonstrating that VUGEN achieves performance comparable to SOTA UMMs of a similar scale. We can move these evaluations to the main body of the paper if the reviewer feels this would improve the positioning of our work.
>
> As for visual reasoning capabilities, we did not report them as they are not the primary focus of our base VLM. It is important to note that VUGEN is (1) agnostic to the base VLM and (2) fully retains the VLM's original understanding and reasoning capabilities thanks to its Mixture-of-Transformers design. Consequently, VUGEN can be seamlessly migrated to a stronger base VLM that possesses more advanced reasoning capabilities.
>
> **Weakness 3: Clarifying the contribution of understanding priors**
>
> The contribution of the understanding priors is demonstrated by the performance improvement of VUGEN—which uses the understanding model's feature space as the target generation space—over our decoupled baseline that uses a distinct vision latent space (specifically, the commonly used SD-VAE latent space). This proves that understanding priors help achieve better generation metrics. We provide qualitative results in Figure 4 and quantitative results in Table 1 and Figure 3.
>
> While we believe the experimental proof strongly demonstrates the method's efficacy, we will also strive to find a more formal explanation to further support these findings in future work.
>
> **Weakness 4: Proving that generation in the understanding latent space is challenging.**
>
> Empirically, Figure 7 (right) shows the difficulty of generation in the understanding space. When trained with identical compute, the model attempting to generate directly in the full understanding space $Z$ (r=1) shows no FID improvement over training, while the generative models learned in the reduced space $\tilde{Z}$ (r=4,8,16,32) train correctly. This divergence is not a small degradation—it is a complete failure to learn the data distribution in $Z$. The empirical gap therefore quantifies the practical tractability difference between the two spaces.
>
> The difficulty stems from the geometry of the underlying representation spaces. The native VLM embeddings in $Z$ are very high-dimensional (1024 in our setting) and not trained for reconstruction or generative smoothness. Our proposed reducer explicitly contracts this space while preserving the relevant information for reconstruction, resulting in a latent distribution with a lower intrinsic dimension and improved local smoothness, which diffusion models can learn effectively—consistent with our empirical observations.

---

> ### Author Response · Authors · 2025-11-20
> **Response to reviewer's questions**
>
> **Weakness 5: Cost-benefit.**
>
> The training cost decomposes into two stages: (1) training the main generation tower in the compressed understanding space, and (2) training the pixel decoder.
>
> For stage (1), the computational cost is comparable to competing approaches that train a generation module on top of a decoupled vision encoder (e.g., JanusFlow, Bagel). VUGEN replaces their generative latent space with our compressed understanding space, which has a similar overall dimension (e.g., SD-VAE tokens: 64x64x16, compressed understanding latent with our default r=16: 32x32x64). Therefore, the training budget is in fact smaller for VUGEN, due to the larger patch size used in understanding latents, reducing the number of tokens (by a factor of 4 in our experiments, and therefore leading to a similar speed-up).
>
> For stage (2), the pixel decoder (PDD) introduces a cost comparable to training a standard VAE decoder, and it remains modest relative to stage (1). Overall, this two-stage process aligns with existing pipelines in terms of computational requirements and scales similarly when increasing model size or dataset scale.

---

> ### Author Response · Authors · 2025-11-26
>
> Dear reviewer 7WAU, we would greatly appreciate your feedback on whether our response has adequately addressed your concerns—if not, please let us know how we can further clarify any remaining points.

---

### Official Review · Reviewer_Htvt · 2025-10-24

**Soundness:** 3
**Presentation:** 3
**Contribution:** 2
**Rating:** 2
**Confidence:** 4

**Summary:**

This paper introduces VUGEN, a unified vision language model for multimodal understanding and generation. To equip a pre-trained VLM with image generation capability, VUGEN transforms the high-dimensional latent space of the VLM's native vision encoder into a lower-dimensional one to simplify VLM's training for generation while preserving visual information.

**Strengths:**

- The paper is well-written and easy to follow.
 - The generation experiments are done on diverse datasets and evaluated with compreshensive metrics.
 - The ablation experiments on dimension reduction ratio provides valuable insights into the trade-off between the generation task and the reconstruction tasks.

**Weaknesses:**

- Reconstruction metrics like PSNR, SSIM, LPIPS, rFID are not reported to compare the proposed pretrained vision encoder + dimension reducer + pixel decoder with existing autoencoders (e.g., SD-VAE and Flux-VAE) used in other diffusion- or autoregressive-based image generation models.
 - The paper only considers two pixel decoder designs, LDM and PDD, both of which are diffusion based models. Another simple but important baseline would be a standard convolutional decoder commonly used in autoencoders like SD-VAE and Flux-VAE.
 - The image understanding task uses the original vision encoder features, while the image generation task uses the compressed representations from the dimension reducer, resulting a gap between the two spaces. What is the advantage of this unified design compared to decoupled vision encoders baselines if semantically aligned VAEs like VA-VAE[1] are used?
 - The baselines reported in Table 1 for ImageNet generation appear relatively weak. The current state-of-the-art FID score on ImageNet 256x256 is below 2. Also, only the SD3 VAE is considered in the decoupled vision encoders baseline. Exploring autoencoders like VA-VAE[1], which incorporate semantic information, would provide a more complete comparison.
 - As VUGEN introduces a separate module for image generation, it would be helpful to clarify its advantages over previous methods like LaVIT[2] and Emu[3] which introduce another diffusion model for image generation based on VLM output features?

[1] Taming Optimization Dilemma in Latent Diffusion Models

[2] Unified Language-Vision Pretraining in LLM with Dynamic Discrete Visual Tokenization

[3] Emu: Generative Pretraining in Multimodality

**Questions:**

Please refer to the weakness session.

---

> ### Author Response · Authors · 2025-11-20
> **Response to reviewer's questions**
>
> We thank the reviewer for their detailed feedback. We address each concern below:
>
> **Weakness 1: Missing quantitative comparison on image reconstruction**
>
> We will provide the results once we finish the experiments.
>
> **Weakness 2: Justifying the choice of Diffusion-based pixel decoders**
>
> Although it is indeed possible to study a larger number of alternative architectures for the pixel decoder, we leave such a study for future work, as the main question we answer in the current paper is to what extent a multimodal generative model can make use of its understanding features to drive its image generation capabilities. That being said, let us further clarify the choice of decoders we studied in our work.
>
> Standard VAE decoders are designed to invert a reconstruction-optimized, low-dimensional latent space that has been explicitly trained to be smooth, dense, and easy to decode. In contrast, VUGEN’s pixel decoder must map from a pretrained VLM’s understanding embedding space, which is high-dimensional, highly structured, and semantically rich. This space is not trained for reconstruction, and its distribution is considerably more complex than VAE-style latents.
>
> As a result, directly decoding VLM embeddings with a simple feedforward convolutional generator is extremely challenging, as the model must learn a much more complex conditional distribution. In practice, generative modeling from such spaces typically relies on diffusion models, which are robust to multimodal and irregular distributions. This is why our investigation focuses on diffusion-based decoders (LDM and PDD).
>
> Furthermore, a feedforward convolutional network would likely require a GAN loss, which is difficult to tune properly to avoid instability. Finally, we note that our proposed PDD, after distillation, effectively becomes a feedforward network for pixel decoding.
>
> **Weakness 3: Advantage of VUGEN’s generation space (compressed understanding representation), compared to semantically aligned VAEs? Comparison with VA-VAE**
>
> VA-VAE trains a VAE tokenizer’s latent representation to be aligned with pre-trained vision foundation model (DINOv2) embeddings. It then trains an image generative model on these semantically aligned tokens. This is conceptually similar to our baseline REPA, which instead aligns intermediate representations during generation with pretrained vision embeddings. Both VA-VAE and REPA *implicitly* align the generation space or process with understanding priors.
>
> In contrast, VUGEN *explicitly* learns to generate within the VLM’s own pretrained embedding manifold, with the dimension reducer serving only as a tool for tractability. Thus, unlike VA-VAE and REPA, our approach does not introduce an additional visual latent space as the generative target. The key advantage is the direct reuse of the VLM’s pretrained visual priors. This leads to improved cross-modal consistency, as the generation process remains grounded in the VLM’s existing semantics. The performance gain in generation is evidenced in Table 1.
>
> We agree that VA-VAE is a valuable additional baseline. For a fair comparison, we would need to (1) train a tokenizer using VA-VAE’s alignment loss with the pretrained VLM’s vision embeddings, and (2) train our generative tower on this aligned token space. Due to rebuttal-phase time and resource constraints, we will include this in future work.

---

> > ### Author Response · Authors · 2025-11-22
> > **Response to reviewer's questions**
> >
> > **Weakness 1: Missing evaluation on pixel decoders**
> >
> > We provide quantitative results on standard reconstruction metrics using ImageNet, comparing our VAE-free pixel decoder against the widely-used SD-VAE tokenizer (which is also the decoder used in our decoupled baseline). The results are as follows:
> >
> > | **Method** | **rFID@50k** | **MSE** | **PSNR** | **SSIM** | **LPIPS** | **dreamsim** |
> > | --- | --- | --- | --- | --- | --- | --- |
> > | **SD-VAE** | 0.591 | 3.614 | 24.42 | 0.758 | 0.0573 | 0.0391 |
> > | **Pixel decoder (PDD)** | 5.467 | 9.080 | 20.42 | 0.624 | 0.1110 | 0.0617 |
> >
> > We observe that our pixel decoder indeed underperforms SD-VAE on image reconstruction. However, we posit that this is an expected outcome, and that this degradation in reconstruction pays significant dividends in the improvement of image generation quality (our main contribution).
> >
> > The SD-VAE latent space is explicitly optimized for high-fidelity reconstruction, as this is its primary training objective. In contrast, our method decodes from a VLM’s understanding space, which is optimized for high-level semantic representation, not pixel-level detail, but it is a superior target space for generation. This can be shown in Table 1 of our main paper, where VUGEN, which generates in the understanding space, achieves considerable improvement in end-to-end generation quality (e.g., 4-point lower FID) compared to the decoupled baseline whose target space is SD-VAE.
> >
> > Meanwhile, the understanding embedding inherently discards low-level texture information that is critical for perfect reconstruction but superfluous for diverse image generation. This can be demonstrated by Figure 5 in the main paper, where generation results from the understanding space show clear enhancement in image generation diversity compared to generation from VAE latents.
> >
> > In summary, while our pixel decoder is less suited for mere reconstruction, this is a deliberate design choice that enables the stronger generative performance that is the central contribution of our work.

---

> ### Author Response · Authors · 2025-11-20
> **Response to reviewer's questions**
>
> **Weakness 4: Baselines on ImageNet FID are too weak.**
>
> We agree with the reviewer that the current state-of-the-art (SOTA) ImageNet-256×256 FID (below 2) is superior to the numbers reported in Table 1. However, these results are achieved by specialized, image-only generative models (e.g., SiT-XL/2+REPA DINOv2 achieves FID=1.42 [1]).
>
> First, VUGEN is not an image-generation expert model but a unified VLM, and our goal is to empower it with generative capability given the constraint of preserving its multimodal understanding ability. As a result, our evaluation compares against other VLM-compatible approaches, not image-only SOTA generators (see our response to reviewer 7WAU).
>
> Second, due to compute constraints, all experiments are conducted with a 1B-parameter VLM. Under this fair comparison within the VLM regime, VUGEN achieves state-of-the-art performance among models of similar size (see Table 2), demonstrating the effectiveness of our proposed method in the intended setting.
>
> **Weakness 5: Advantage compared to LaVIT and EMU?**
>
> VUGEN differs from LaVIT and Emu in its objective and workflow. LaVIT and Emu perform joint pre-training for image understanding and generation. Their models start from EVA-CLIP and introduce new “image token” representations (continuous causal 1D for Emu, discrete for LaVIT), and then train a diffusion model on these new tokens for pixel-level generation. In contrast, VUGEN starts from an already-trained VLM with established visual understanding abilities. Our goal is to add high-quality image generation without disrupting its existing capabilities, making the settings not directly comparable.
>
> VUGEN demonstrates two key advantages:
>
> First, the image embeddings from EVA-CLIP lack low-level visual detail, causing diffusion models trained on them to struggle with fine-grained textures and spatial structure (see Figure 7 in LaVIT). VUGEN generates directly within the VLM's native understanding feature space, and our dimension reducer is explicitly trained to preserve both semantic and low-level information. Empirically, our reconstructions retain significantly more detail (Figures 2 and 6), demonstrating richer visual fidelity.
>
> Second, both LaVIT and Emu depend on a latent diffusion model that itself uses an additional VAE tokenizer, adding architectural complexity and reconstruction bias. In contrast, VUGEN introduces PDD, a VAE-free pixel diffusion decoder, eliminating the external autoencoder, simplifying the pipeline, and creating a more direct connection between the VLM's visual space and the generated pixels.
>
> [1] Yu, Sihyun, et al. "Representation Alignment for Generation: Training Diffusion Transformers Is Easier Than You Think." The Thirteenth International Conference on Learning Representations.

---

> ### Author Response · Authors · 2025-11-26
>
> Dear reviewer Htvt, we would greatly appreciate your feedback on whether our response has adequately addressed your concerns—if not, please let us know how we can further clarify any remaining points.

---

### Official Review · Reviewer_1rBA · 2025-10-28

**Soundness:** 3
**Presentation:** 3
**Contribution:** 3
**Rating:** 6
**Confidence:** 4

**Summary:**

The paper proposes VUGEN, a two-stage approach to equip a unified VLM with image generation by directly leveraging its native visual understanding features. Stage 1 learns a dimension reducer g that projects the high-dimensional understanding embeddings z from the VLM’s vision encoder into a reduced, tractable space ˜Z optimized jointly with a pixel decoder d (either a finetuned LDM or a lightweight pixel-space diffusion decoder, PDD). Stage 2 freezes g and trains a generative head (Mixture-of-Transformers tower) via rectified flow matching to sample ˜z ∼ P(˜z|c), then decodes to pixels x via d(˜z). The key idea is to align generation with the model’s understanding priors, avoiding representation mismatch from separate (VQ-)VAE tokenizers and the complexity of bridging to external diffusion models.

  Empirically, VUGEN improves prompt-following and fidelity on COCO (models trained on StockMix): DPG-Bench 71.17→74.32 and FID 11.86→9.06 vs. a REPA-aligned decoupled baseline; and outperforms baselines trained on ImageNet (FID 5.40→4.15, Density/Coverage up). Ablations show: (i) directly generating in Z is hard; a jointly learned reducer outperforms PCA; (ii) pixel-space diffusion decoder (PDD) achieves comparable reconstructions to LDM but with far fewer params (48M vs. 794M) and higher throughput; (iii) a reduction ratio r≈16 balances generative tractability and decoding difficulty. Understanding performance is preserved at base VLM levels (Table 3).

**Strengths:**

- Clear, well-motivated design: samples in a reduced version of the VLM’s understanding space, preserving alignment between understanding and generation; strong rationale and ablations (PCA vs. learned reducer; r trade-off).
- Competitive results with careful baselines sharing architecture/data/training: improves FID/DPG/GenEval across StockMix→COCO and ImageNet settings; analysis over guidance scale clarifies realism–consistency trade-offs.
- Practical decoder findings: pixel-space diffusion decoder rivals LDM while being far smaller and faster; avoids dependence on VAE latents, reducing complexity.
- Preserves understanding: retains PLM-1B’s comprehension performance; shows a path to unified MLLMs without decoupled vision tokenizers.

**Weaknesses:**

- Data provenance and comparability: the main training uses a mixed StockMix (YFCC100M, CC12M, and a proprietary S320M recaptioned with Florence-2). While baselines share this setup, cross-paper comparability is limited; clearer licensing/availability statements for S320M would help.
- Limited scope of understanding preservation: while Table 3 suggests parity on standard benchmarks, it would be useful to test for regressions in more fine-grained or long-context visual reasoning after generative training, especially under higher r.
- Generative scaling and distribution shift: results are at 256×256; how do trends hold at higher resolution and for out-of-domain prompts? Also, how stable is training when swapping in different base VLM encoders (e.g., DINOv2 or SigLIP variants)?
- Theoretical underpinnings: the choice of rectified flow matching is reasonable; adding a brief justification vs. diffusion loss and showing a small apples-to-apples comparison would strengthen claims of sample efficiency.
- Decoder choice vs. alignment: PDD and LDM are “similar” in reconstructions; however, prompt alignment (DPG/GenEval) contributions per component (reducer vs. generator vs. decoder) could be clarified via controlled ablations.

**Questions:**

- Does training the reducer jointly with PDD ever reduce understanding robustness? Can you report pre/post shifts on more challenging understanding tasks (e.g., MMMU categories requiring fine localization)?
- How sensitive are results to r and g’s architecture across datasets? Is there a principled way (e.g., information bottleneck or spectral metrics) to set r per vision encoder?
- Could you show a small table isolating “generate-in-Z” vs. “generate-in-˜Z” under the same compute, to quantify the tractability gap, beyond anecdotal FID>200?
- For external comparability: do you have COCO metrics when training only on public data (e.g., YFCC100M+CC12M without S320M), to contextualize gains relative to models trained purely on public datasets?
- PDD details: you mention distillation to a single-step decoder; can you quantify speedups at sample time for end-to-end T2I, not just reconstruction throughput?

---

> ### Author Response · Authors · 2025-11-20
> **Response to reviewer's questions**
>
> We thank the reviewer for their detailed feedback. We address each concern below:
>
> **Weakness 1: Clarification on data provenance and license**
>
> VUGEN, as well as our baselines, were all trained using the same StockMix data mix consisting of CC12M, YFCC100M, and S320. The latter dataset consists of 320M licensed Shutterstock images and captions. We included this licensed dataset in our training mix to increase the size of our training set 4x. Other public dataset alternatives come with associated risks and limitations regarding data use. Although we agree that the use of non-standard data mixes precludes some direct comparisons to prior work, we feel that controlled experiments to compare with baselines strike the right balance between responsible data use and training models at a reasonable scale.
>
> **Weakness 2: Missing visual understanding and reasoning evaluation.**
>
> We did not initially include evaluation on fine-grained or long-context visual reasoning as this is not the primary focus of our base VLM. However, it is important to note that VUGEN is (1) agnostic to the base VLM and (2) fully retains the VLM's original understanding and reasoning capabilities thanks to its Mixture-of-Transformers [1] design. Consequently, VUGEN can be seamlessly migrated to a stronger base VLM that possesses more advanced reasoning capabilities.
>
>
> **Weakness 3: Scalability and generalisability**
>
> **Scaling to higher resolutions.**
> VUGEN can scale to higher-resolution generation in two ways:
> - **Upgrading the base VLM.** Since VUGEN operates entirely on the VLM’s native vision embeddings, it is agnostic to the specific VLM and its vision encoder. The methodology is therefore compatible with VLMs equipped with pretrained vision encoders such as DINOv2 or SigLIP variants. In principle, swapping in VLMs with higher-resolution native encoders directly increases the effective resolution limit. We did not run full experiments with alternative VLMs during the rebuttal period due to compute constraints, but the methodology is fully compatible.
> - **Training a higher-resolution pixel decoder.** Our proposed PDD pixel decoder can be trained to map the same fixed VLM features to higher-resolution images. This allows for seamless adaptation from the same generative tower without modifying the VLM itself. We plan to include this exploration in future work.
>
> **Generalization to out-of-domain prompts.**
> VUGEN inherits the semantic prior of the underlying VLM. Prompt generalization capability is measured through DPG-Bench, which contains challenging compositional, multi-object, long-prompt, and attribute-heavy instructions. VUGEN achieves performance comparable to larger SOTA models (Table 2), indicating robustness to semantic and prompt distribution shift.
>
> **Weakness 4: More justification on choice of flow matching**
>
> VUGEN adopts rectified flow matching rather than a classic diffusion loss (e.g., DDPM) because flow matching addresses the primary weakness of DDPMs—slow sampling. The DDPM process is inherently curved and stochastic, forcing the generative model to navigate a complex trajectory that requires many small iterative steps to reverse. The rectified flow paradigm reformulates this as a straight-line, deterministic flow path, which is both simpler to learn and more efficient to simulate. We invite the reviewer to refer to the SD3 paper [2] for more details. Typically, we use a sampling step count of 6 for our flow matching, as opposed to the typical 50 or 100 steps for diffusion sampling.

---

> ### Author Response · Authors · 2025-11-20
> **Response to reviewer's questions**
>
> **Question 1: Does joint training of the reducer + PDD affect understanding robustness?**
>
> No. VUGEN’s design ensures that the reducer and PDD do not affect the VLM’s understanding ability.
> - Understanding inference always uses the original, uncompressed vision embeddings produced by the frozen pretrained vision encoder; the reducer is never involved at inference time for understanding tasks. (
> - During training, we follow the standard Mixture-of-Transformer [1] paradigm: the understanding tower of the VLM is frozen and only the generation branch is updated.
>
> Thus, the reducer does not change the VLM’s perception features; it only creates a more tractable latent space for the generation task. Because the core VLM encoder is untouched, its robustness remains unchanged.
>
> **Question 2: Sensitivity to the reducer ratio ($r$) and architecture ($g$); is there a principled way to choose ($r$)?**
>
> Figure 7 illustrates the effect of varying the dimension-reduction ratio ($r$): increasing $r$ (more compression) reduces reconstruction fidelity (Fig. 7, left) but improves generative modeling tractability, resulting in better generation FID (Fig. 7, right).
> We appreciate the reviewer’s suggestion regarding principled criteria. Inspired by this, we propose selecting $r$ by examining the reconstruction–generation tradeoff curve (reconstruction FID vs. generation FID) and choosing the “elbow point,” where small decreases in reconstruction quality begin to yield sharp improvements in generation performance. This provides a practical and data-driven method for setting $r$ per vision encoder.
> We have not yet performed an ablation on the reducer architecture ($g$), but we agree it is an important factor and will include a systematic study in future work.
>
> **Question 3: Comparison of generating in ($Z$) vs. generating in ($\tilde{Z}$) under matched compute**
>
> Figure 7 (right) presents a direct comparison under equal compute: the model trained to generate directly in the full VLM embedding space ($Z$), shown in cyan, fails to make progress and the FID remains extremely high throughout training. In contrast, generation in the reduced space ($\tilde{Z}$), shown in green, converges steadily.
> This gap quantifies the fundamental tractability advantage of operating in $\tilde{Z}$ The failure of the model trained in $Z$ to converge with identical compute illustrates that the original VLM embedding distribution is too complex to model directly, validating the motivation for the reducer.
>
> **Question 4: Contextualizing gains with public data**
>
> We acknowledge the reviewer's valid point regarding the importance of comparison on public data. Unfortunately, due to time constraints during the rebuttal phase, we were unable to complete the full two-stage training cycle using only public datasets (YFCC100M+CC12M). We will include it as an important direction for future work.
>
> **Question 5: Quantifying end-to-end sampling speedup**
>
> We will provide the quantified results once we finish the evaluation.
>
> [1] Liang, Weixin, et al. "Mixture-of-Transformers: A Sparse and Scalable Architecture for Multi-Modal Foundation Models." ICLR 2025 Workshop on World Models: Understanding, Modelling and Scaling.
>
> [2] Esser, Patrick, et al. "Scaling rectified flow transformers for high-resolution image synthesis." Forty-first international conference on machine learning. 2024.

---

> > ### Author Response · Authors · 2025-11-22
> > **Response to reviewer's questions**
> >
> > **Question 5: Quantifying end-to-end sampling speedup**
> >
> > We provide a detailed analysis of the inference time for end-to-end text-to-image generation. We evaluate VUGEN with both proposed pixel decoders, as well as our decoupled baseline (which uses SD-VAE). This evaluation was performed on H200 GPUs with a batch size of 32.
> >
> > | Method | Generation of understanding latents  (s/latent) | Decoding to pixel images (s/image) | Total (s/image) |
> > | --- | --- | --- | --- |
> > | Decoupled baseline | 0.3360 | 0.0096 | 0.3456 |
> > | VUGEN (LDM) | 0.3378 | 0.6733 | 1.0111 |
> > | VUGEN (PDD) | 0.3360 | 0.0828 | 0.4188 |
> >
> > The sampling process can be decomposed into two parts. The first part is the generation of understanding latents from the image generation tower, which remains nearly identical for all methods. The difference arises in the second part—the decoding of latents into images by the pixel decoder. Crucially, our proposed PDD decoder achieves a sampling speed similar to that of the VAE. Regarding the details of PDD's distillation process, this is not the primary focus of our paper. We kindly refer the reviewer to the SSDD submission [3] for a comprehensive explanation.
> >
> > [3] Anonymous. SSDD: Single-step diffusion decoder for efficient image tokenization. In ICLR, 2026.
> > URL https://openreview.net/forum?id=G1o4HNtOLO. Under anonymous submission.

---

> ### Author Response · Authors · 2025-11-26
>
> Dear reviewer 1rBA, we would greatly appreciate your feedback on whether our response has adequately addressed your concerns—if not, please let us know how we can further clarify any remaining points.

---

### Official Review · Reviewer_AuJA · 2025-11-01

**Soundness:** 3
**Presentation:** 3
**Contribution:** 3
**Rating:** 6
**Confidence:** 5

**Summary:**

The authors propose leveraging visual understanding priors for both visual perception and generation tasks. They transform a high-dimensional semantic latent space into a low-dimensional, tractable distribution that preserves essential visual information. A pixel diffusion model is then trained to generate images from these latent representations. Experimental results demonstrate that, by utilizing a unified semantic visual representation, the method achieves superior image generation performance.

**Strengths:**

1. The proposed method is intriguing and demonstrates that generating semantic visual latent features can lead to improved image generation performance.
2. To make the generation process feasible, the authors introduce a dimension reduction module, which is a simple yet effective design.
3. The paper is well-written and easy to follow.

**Weaknesses:**

1. There is no quantitative comparison between the proposed pixel decoder and other mainstream tokenizers, such as VAE and latent diffusion decoders. The authors should include relevant metrics (e.g., rFID) to assess the performance. It remains unclear how well the proposed pixel decoder performs. If its results are significantly worse, it would suggest substantial information loss when generating images from the semantic latent space.
2. The image generation module in the mixture-of-transformers contains only 0.2B parameters. This relatively small capacity raises concerns about the reliability and persuasiveness of the results. I encourage the authors to increase the model size for the image generation component to validate the scalability and robustness of the approach.
3. The dataset S320M is not widely adopted in the community, particularly for unified understanding and generation tasks. The authors should clarify their motivation for using this dataset. Furthermore, the use of such datasets may contribute to the relatively weaker performance on more modern benchmarks, such as GenEval and DPGBench, which diminishes the overall persuasiveness of the work.

**Questions:**

Please refer to the Weakness section.

---

> ### Author Response · Authors · 2025-11-20
> **Response to reviewer's questions**
>
> We thank the reviewer for their detailed feedback. We address each concern below:
>
> **Weakness 1: Missing evaluation on pixel decoders**
>
> We will provide the required evaluation results on pixel decoders once we finish the experiments.
>
> **Weakness 2: Scalability**
>
> We first clarify a misunderstanding regarding the parameter count. Under the Mixture-of-Transformers (MoT) setup, we start from a 1B-parameter pretrained VLM (PLM-1B). For the generation branch, we initialize a full copy of the transformer tower from the same pretrained weights and add the vision connector and projection layers required for generation, while keeping the original understanding tower frozen.
> As a result, the total trainable parameters for image generation is ~1.2B, not 0.2B. The 0.2B number refers only to the newly added modules (reducer, connectors, and PDD), but the generation transformer itself contributes ~1B trainable parameters. Therefore, the effective capacity of the generation module is on par with most comparable works evaluating multimodal generative extensions at the 1B scale.
> We agree that scaling to larger VLM backbones (e.g., 7B) is a valuable direction. However, due to compute limitations during the rebuttal period, we are unable to conduct full-scale additional training runs. We will include larger-scale experiments as part of future research.
>
> **Weakness 3: Clarification on data provenance and license**
>
> VUGEN, as well as our baselines, were all trained using the same StockMix data mix consisting of CC12M, YFCC100M, and S320. The latter dataset consists of 320M licensed Shutterstock images and captions. We included this licensed dataset in our training mix to increase the size of our training set 4x. Other public dataset alternatives come with associated risks and limitations regarding data use. Although we agree that the use of non-standard data mixes precludes some direct comparisons to prior work, we feel that controlled experiments to compare with baselines strike the right balance between responsible data use and training models at a reasonable scale.

---

> > ### Author Response · Authors · 2025-11-22
> > **Response to reviewer's questions**
> >
> > **Weakness 1: Missing evaluation on pixel decoders**
> >
> > We provide quantitative results on standard reconstruction metrics using ImageNet, comparing our VAE-free pixel decoder against the widely-used SD-VAE tokenizer (which is also the decoder used in our decoupled baseline). The results are as follows:
> >
> > | **Method** | **rFID@50k** | **MSE** | **PSNR** | **SSIM** | **LPIPS** | **dreamsim** |
> > | --- | --- | --- | --- | --- | --- | --- |
> > | **SD-VAE** | 0.591 | 3.614 | 24.42 | 0.758 | 0.0573 | 0.0391 |
> > | **Pixel decoder (PDD)** | 5.467 | 9.080 | 20.42 | 0.624 | 0.1110 | 0.0617 |
> >
> > We observe that our pixel decoder indeed underperforms SD-VAE on image reconstruction. However, we posit that this is an expected outcome, and that this degradation in reconstruction pays significant dividends in the improvement of image generation quality (our main contribution).
> >
> > The SD-VAE latent space is explicitly optimized for high-fidelity reconstruction, as this is its primary training objective. In contrast, our method decodes from a VLM’s understanding space, which is optimized for high-level semantic representation, not pixel-level detail, but it is a superior target space for generation. This can be shown in Table 1 of our main paper, where VUGEN, which generates in the understanding space, achieves considerable improvement in end-to-end generation quality (e.g., 4-point lower FID) compared to the decoupled baseline whose target space is SD-VAE.
> >
> > Meanwhile, the understanding embedding inherently discards low-level texture information that is critical for perfect reconstruction but superfluous for diverse image generation. This can be demonstrated by Figure 5 in the main paper, where generation results from the understanding space show clear enhancement in image generation diversity compared to generation from VAE latents.
> >
> > In summary, while our pixel decoder is less suited for mere reconstruction, this is a deliberate design choice that enables the stronger generative performance that is the central contribution of our work.

---

> ### Author Response · Authors · 2025-11-26
>
> Dear reviewer AuJA, we would greatly appreciate your feedback on whether our response has adequately addressed your concerns—if not, please let us know how we can further clarify any remaining points.

---

### Meta-Review · Area_Chair_a7gc · 2026-01-06

**Summary:**

The paper proposes VUGEN, a framework that uses the pretrained visual understanding prior for image generation. This paper received a mixed rating from the reviewers, with ratings of 6, 6, 2, and 2. Reviewers Htvt and 7WAU, with a rating of 2, raised concerns regarding the absence of reconstruction metrics, the limited comparison with different decoders, weak image generation performance, and the fundamental motivation.

After a careful review of the paper, the reviewer comments, and the rebuttal, the AC has decided to reject this submission. Although the authors provided responses during the rebuttal period, the major issues concerning the motivation and the suboptimal image generation performance remain unaddressed. The AC concludes that the work does not currently demonstrate a sufficient performance advantage or theoretical justification for acceptance.

**Reviewer Concerns:**

The concerns raised by Reviewers Htvt and 7WAU regarding the suboptimal image generation performance and the fundamental motivation for the proposed method's advantages have not been fully addressed.

**Reviewer Scores:**

Given that the major concerns raised by Reviewers Htvt and 7WAU have not been fully addressed, it is unlikely that their scores will improve.

---

### Decision · Program_Chairs · 2026-01-26

Reject